# A Multiscale Attention Segment Network-Based Semantic Segmentation Model for Landslide Remote Sensing Images

Nan Zhou [1,2,3], Jin Hong [1,2,3], Wenyu Cui [1,3], Shichao Wu [1,3] and Ziheng Zhang [4,*]

1 Anhui Institute of Optics and Fine Mechanics, Hefei Institutes of Physical Science, Chinese Academy of Sciences, Hefei 230031, China; nanchou@mail.ustc.edu.cn (N.Z.); hongjin@aiofm.ac.cn (J.H.); cuiwenyu@aiofm.ac.cn (W.C.); wusc@aiofm.ac.cn (S.W.)
2 University of Science and Technology of China, Hefei 230026, China
3 Key Laboratory of General Optical Calibration and Characterization Technology, Hefei Institutes of Physical Science, Chinese Academy of Sciences, Hefei 230031, China
4 Hefei Institutes of Physical Science, Chinese Academy of Sciences, Hefei 230031, China
* Correspondence: zhzhang@inest.cas.cn

**Abstract:** Landslide disasters have garnered significant attention due to their extensive devastating impact, leading to a growing emphasis on the prompt and precise identification and detection of landslides as a prominent area of research. Previous research has primarily relied on human–computer interactions and visual interpretation from remote sensing to identify landslides. However, these methods are time-consuming, labor-intensive, subjective, and have a low level of accuracy in extracting data. An essential task in deep learning, semantic segmentation, has been crucial to automated remote sensing image recognition tasks because of its end-to-end pixel-level classification capability. In this study, to mitigate the disadvantages of existing landslide detection methods, we propose a multiscale attention segment network (MsASNet) that acquires different scales of remote sensing image features, designs an encoder–decoder structure to strengthen the landslide boundary, and combines the channel attention mechanism to strengthen the feature extraction capability. The MsASNet model exhibited an average accuracy of 95.13% on the test set from Bijie's landslide dataset, a mean accuracy of 91.45% on the test set from Chongqing's landslide dataset, and a mean accuracy of 90.17% on the test set from Tianshui's landslide dataset, signifying its ability to extract landslide information efficiently and accurately in real time. Our proposed model may be used in efforts toward the prevention and control of geological disasters.

**Keywords:** deep learning; remote sensing images; landslide identification; semantic segmentation; feature extraction

## 1. Introduction

A landslide is a natural phenomenon in which the soil or rock on a slope is influenced by factors such as erosion of rivers, groundwater, rainwater soaking, earthquakes, and artificial slope cutting, and slides downslope along a certain weak surface under gravity. Landslides rank as the second most significant geological peril, following earthquakes. These phenomena exhibit a global distribution and manifest with regularity on an annual basis, resulting in substantial financial losses and a significant loss of human lives [1]. According to the 2022 Statistical Bulletin issued by the Ministry of Natural Resources of China, 5659 geological disasters occurred in China that year, including 3919 landslides, 1366 avalanches, 202 mudslides, 153 ground collapses, 4 ground cracks, and 15 ground subsidence events. A total of 90 people were killed, 16 went missing, 34 were injured, and the direct economic loss was 1.50 billion yuan (https://www.mnr.gov.cn/ accessed on 6 March 2024). Landslides are the most prominent type of geological disaster in China, and as a result of global climate change, seismic activity, and accelerated development, landslides are becoming more frequent, and their real dangers and potential hazards are

becoming more evident. Hence, it is important to conduct landslide detection in a timely manner following the occurrence of a landslide to ascertain its area, scale, and distribution; this information aids disaster mitigation and relief efforts and can further assist planning and construction activities in affected areas.

Traditional landslide monitoring methods are primarily based on manual field surveys, which have low efficiency and poor real-time performance. As Earth observation technology evolves, the significance of remote sensing in identification has grown, providing early warnings for large-scale natural disasters due to its timeliness, extensive coverage area, and detailed information. Therefore, the rapid and accurate identification of landslide disaster impact ranges from massive amounts of remote sensing image data is integral in guiding disaster prevention and mitigation. Compared to traditional methods, remote sensing technology has the advantages of large-scale synchronous observations, real-time operation, and low cost [2], making it suitable for the rapid and continuous monitoring of slopes. Remote sensing data that are widely used for landslide monitoring can be divided into high- and medium-resolution remote sensing images. Among them, medium-resolution images contain rich spectral information and are widely used in large-scale interpretation scenarios owing to their large width and relatively low cost; however, owing to the constraints of spatial resolution, it is difficult to accurately extract the shape, edges, number of slopes, and other detailed information from these images. High-resolution images can provide finely detailed information on the shape of landslides, boundaries, and other detailed information. These advantages are in accordance with the development needs of disaster prevention and mitigation, meaning that high-resolution images have an irreplaceable advantage in the accurate investigation of landslides in key areas.

Spectral, textural, and geometric features that traditional classification methods rely on comprise manually extracted low-level features, which are highly subjective, insufficiently expressive, and difficult to apply to areas with complex backgrounds and obscure slope features. With the rapid development of deep learning technology in the field of image processing, semantic segmentation has gradually become the mainstream method of image pixel-level decoding [3], the purpose of which is to annotate each pixel in the image with a category label; this is consistent with the purpose of remote sensing pixel-level classification. Semantic segmentation deep learning methods are end-to-end algorithms that can classify segmented pixels by directly inputting image information as a supervised signal [4]. The signal is capable of autonomously identifying key features for a specific task, utilizing a robust loss function [5,6], and manual selection of segmentation parameters or combining multidimensional features is not necessary. Despite the complexity of current deep learning models, their internal decision-making processes remain unclear or even incomprehensible, resembling a "black box". The benefit of using an end-to-end learning approach is the simplicity of achieving a universally optimal solution. Thus, in utilizing the deep learning technique of semantic segmentation, it is possible to classify remote sensing images with high efficiency and accuracy [7]. This technique offers efficient, instantaneous, and highly accurate categorization outcomes for landslide detection.

Accordingly, semantic segmentation is widely used in pixel-level classification tasks of remote sensing images, including building [8], road [9], and cropland extractions [10], as well as surface single-element extractions and surface full-element classifications [11]. Compared with traditional machine learning methods, the main advantage of deep learning is that it can automatically learn multilevel and multiscale features from low to high levels through end-to-end networks [12], and its powerful feature learning capability proves its efficacy in improving the interpretation accuracy of remote sensing images. Landslides have different morphologies owing to the uncertainty of their occurrence time, location, environment, and other factors; therefore, it is difficult to construct a universal landslide recognition model. There is a need for automatic approaches that utilize deep learning algorithms to accurately identify the site of landslides using remote sensing photos for practical purposes. An end-to-end segmentation network known as deep residual shrinkage U-Net (DRs-UNet) was developed by Ma and Mei to extract possible active landslides

from Synthetic Aperture Radar Interferometry (InSAR) data. The network concept was also used for the purpose of detecting landslides in a designated test location situated in Zhongxinrong County, namely along the Jinsha River. The experimental findings provided evidence of the effectiveness of the system in automatically identifying potential landslide threats [13]. In addition, Choi et al. have developed a deep learning framework to digitalize flow-type landslides' kinematics using a fully convolutional neural network (FCNN). The model outperforms traditional algorithms and can process images from consumer-grade cameras under complex conditions [14]. Nugraha et al. proposed a framework for improving the performance of Deep Convolutional Networks (DCN) in aerial imagery semantic segmentation of natural disaster-affected areas. It uses U-Net and Pyramid Scene Parsing Network models, with the Grid Search algorithm improving performance [15].

Deep learning methods autonomously extract the features of the target object through sample training, bypassing the subjectivity of manually constructing features. Moreover, they can form more abstract and stable features of the target object, greatly improving the recognition accuracy. Based on this feature, researchers have begun using deep learning for landslide recognition. Semantic segmentation, an important task in deep learning, has been useful in automated image recognition because of its end-to-end pixel-level segmentation capability. SegNet, proposed by Badrinarayanan [16] et al. and based on U-Net, introduces pooled index links on top of the encoding-decoding structure to achieve efficient and accurate target segmentation. PSPNet (Pyramid Scene Parsing Network), proposed by Zhao et al. [17], uses the pyramid pooling structure to fuse four scale features to enable the network to learn semantic features at different scales. Additionally, the DeeplabV3+ model proposed by Chen et al. [18] incorporates contextual information on top of PSP-Net to improve the edge clarity of segmented targets. Because of the pixel-level classification features of semantic segmentation, this model has a strong potential for landslide hazard recognition based on remote sensing images, and several scholars have conducted in-depth research in this field. Among them, Cheng et al. [19] designed a YOLO SA landslide detection model based on high-resolution remote sensing images based on YOLOV4 and combining Gconv, Gbneck, and an attention mechanism; however, the target detection method cannot accurately identify the landslide boundary information, which is obviously not conducive to post-disaster assessment and damage determination. Ullo et al. [20] used Mask R-CNN to utilize remote sensing images to segment landslides and obtain landslide boundary information. However, when using deep ResNet as the backbone network, the boundaries appear to display an obvious blurring problem. Thus, to obtain clearer landslide boundaries, Bragagnolo et al. [21] used the coding and decoding ability of U-Net to effectively restore the boundary information. Nevertheless, the accuracy of small-scale landslide recognition in remote sensing images is compromised by the disparity in feature scale. Yi et al. [22] designed the LandsNet model for landslide hazard identification by combining residual blocks, attention modules, and multi-scale fusion operation and reached a high identification accuracy; however, the model's complexity and computational cost are relatively high due to the combination of excessive modules. Wan et al. [23] propose an improved deeplabV3+ method that combines BotNet and ResNet feature maps. The experimental results indicate that when this method performs at its peak on the validation set, the average intersection-over-union on the test set is 82.50%. Sreelakshmi [24] has developed an innovative deep-learning framework that uses visual saliency for automatic landslide identification, achieving 94% accuracy, surpassing existing models, and offering a promising tool for risk assessment and management in landslide-prone areas. Li et al. [25] suggest integrating BotNet and ResNet feature maps to create an enhanced deeplabV3+ landslide identification technique. The modified YOLOv8 model can perform well on the validation set, according to experimental data, and it achieves excellent accuracy. Chen et al. [26] propose a squeeze-and-excitation network (SENet) to U-Net for accurate landslide extraction. The model, trained using Sentinel-2A pictures, beats the U-Net and UNet Backbone models with an F1 value of 87.94%, resulting in less false detection and more accurate findings.

To address the disadvantages of the existing remote sensing image-based semantic segmentation model for landslide hazards, which include a fuzzy identification of landslide boundary regions and differentiated feature extraction accuracy of remote sensing images, this study proposes an MsASNet network model. The model acquires remote sensing image features at different scales, designs an encoder–decoder structure to strengthen the landslide boundary, and combines the channel attention mechanism to strengthen the feature extraction capability.

## 2. Materials and Methods

### 2.1. Materials

In this study, we employed three distinct datasets to validate the performance of the algorithm. Ablation studies were conducted using Bijie's landslide dataset to assess the impact of each component within the proposed network on its overall performance. Simultaneously, comparative experiments were carried out using Chongqing's landslide dataset and Tianshui's landslide dataset to further evaluate the performance of the proposed algorithm in segmentation across diverse datasets. The utilization and spread of this methodology have had a beneficial impact.

- Bijie's landslide dataset.

The training dataset of landslides in Bijie City, Guizhou Province, China, used in this experiment is an open-source dataset consisting of 770 landslide samples and 2003 non-landslide samples obtained by Prof. Shunping Ji's team at Wuhan University (http://gpcv.whu.edu.cn/data/Bijie_pages.html accessed on 6 March 2024). These data are based on historical landslide cataloged data, on-site investigations, and a combination of TripleSat remote sensing imagery data. Moreover, the dataset includes landslide images as well as corresponding DEM data and landslide shape masks; the RGB image has a ground resolution of 0.8 m, while the Digital Elevation Model (DEM) has an elevation accuracy of 2 m. The shape vector of each landslide is manually outlined using ArcGIS. The study area is situated in the sloping region of the shift from the Tibetan Plateau to the eastern hills, the research zone features altitudes ranging from 45 to 2900 m above sea level, a significant elevation variance, multiple steep inclines, copious rainfall (with an annual average of 849–1399 mm), and a delicate ecological setting. Bijie City is located on the Yunnan–Guizhou Plateau, which is at the intersection of two or three terraces, and a large relative difference exists in elevation within the region.

Due to the presence of images with varying dimensions in the dataset, it was imperative to standardize their dimensions for input into the neural network for training purposes. For images larger than these specified dimensions, we employed a resizing technique that maintained the aspect ratio, thereby ensuring that the image dimensions conformed to the network's input requirements without distortion. For images larger than 512 × 512 pixels, the resizing process involves reducing the longer side to 512 pixels while proportionally scaling down the shorter side. Subsequently, the resized image is centered and zero-value pixels are utilized to fill in the remaining areas, ensuring the final dimensions of the image are 512 × 512 pixels. Conversely, for images smaller than the aforementioned dimensions, a padding strategy was employed wherein the original image was centered, and zero-value pixels were appended around its periphery until the image dimensions met the specified criteria.

- Chongqing's landslide dataset

Comparative experiments were conducted using Google Earth satellite landslide images from the mountainous areas of Wulong County (107°14′~108°05′E, 29°02′~29°40′N), Chongqing City. The RGB image has a ground resolution of 1 m, while the Digital Elevation Model (DEM) has an elevation accuracy of 5 m. The training dataset of landslides in Wulong County used in this experiment consists of 765 landslide samples. Wulong District is part of Chongqing Municipality and is located in the southeast of the city, in the lower reaches of the Wujiang River and the canyon area of the Wuling and Dalou Mountains

in the southeast of Chongqing. The chosen research site exhibits a subtropical humid monsoon environment characterized by an average relative humidity ranging from 70% to 80% and an average annual precipitation of 1000–1350 mm. This precipitation is primarily concentrated during the flood season spanning from May to September, often accompanied by severe rainstorms. The region has a high topography in the eastern part and a low topography in the western part, with the maximum altitude reaching 2786 m and the lowest altitude reaching 52 m. The geological composition exhibits a complex and varied structure, with a predominant topography characterized by hills and mountains. Landslide disasters are a common occurrence due to the unique characteristics of the topography, geology, and hydrological environment.

- Tianshui's landslide dataset

Comparative experiments were conducted using Google Earth satellite landslide images from the mountainous areas of Tianshui City (104°35′~106°44′E, 34°05′~35°10′N), Gansu Province. The RGB image has a ground resolution of 1 m, while the Digital Elevation Model (DEM) has an elevation accuracy of 5 m. The training dataset of landslides in Tianshui City used in this experiment consists of 197 landslide samples. Tianshui City is characterized by an extensive mountain range that stretches from the northwest to the southeast. The mountains have varying elevations, ranging from 1000 to 2100 m above sea level. Tianshui City has a clear separation in its geomorphology, with the eastern and southern regions of the city being elevated as a result of the presence of ancient stratigraphic folds. This has led to the formation of rough hilly terrain. The northern region experiences geological subsidence and the accumulation of red and yellow soil layers, resulting in the formation of loess deposits and loess hill landforms.

Tianshui City has a temperate monsoon climate with an average yearly temperature of 11 °C and a frost-free period of about 185 days. The average annual precipitation is 574 mm, with a progressive decrease from the southeast to the northwest. Landslide disasters primarily occur in loess hilly regions, affecting all counties and districts within the city, especially Qinzhou District, Maiji District, Wushan County, and Qingshui County, where they are particularly prevalent. The interface between the loess layer and the bedrock beneath it is structurally fragile in these regions. The hills are steep, and the soil is loose as a result of seismic waves and impacts. Additionally, the increased rainfall during the primary flood season greatly facilitates the occurrence of landslide disasters.

The dataset was annotated by several geotechnical experts through on-site investigations and visual interpretation and then screened. The image size within this dataset ranges from 133 × 133 to 256 × 256. The landslide sites of the three selected study areas are shown in Figure 1. The figure represents the areas of Bijie City, Guizhou Province, Tianshui City, Gansu Province, and the mountainous area of Wulong County, Chongqing City, including the DEM data and the distribution of landslide sites.

As shown in Figure 2, we show some landslide images in the test set. The region enclosed by the yellow curve represents the landslide area that has been determined through visual interpretation or field research. During the training process, we train each of the three datasets separately. We randomly divide each dataset into three sets: a training set, a validation set, and a test set. The ratio of the division is 8:1:1. The three datasets are trained independently, with each dataset having its own training set (80%), validation set (10%), and test set (10%).

**Figure 1.** Distribution of landslide sites in the study area.

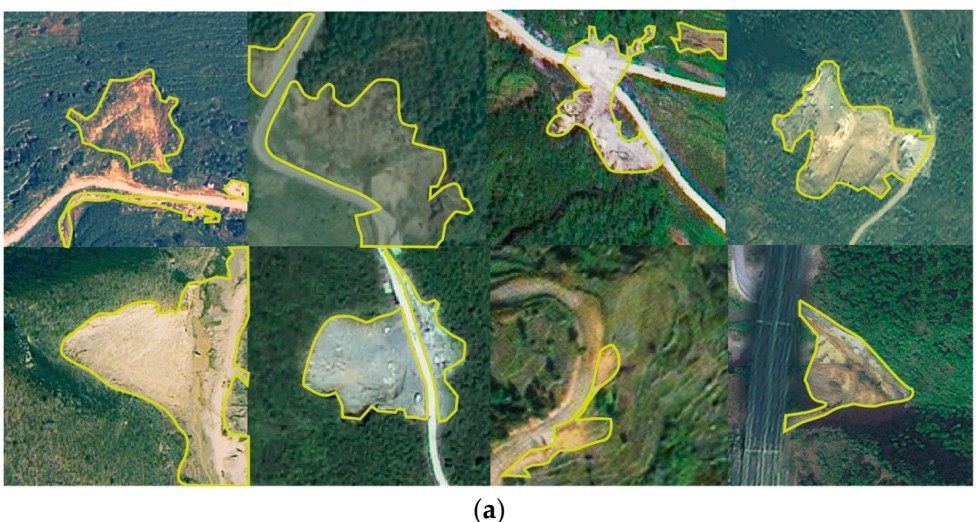

(**a**)

**Figure 2.** *Cont.*

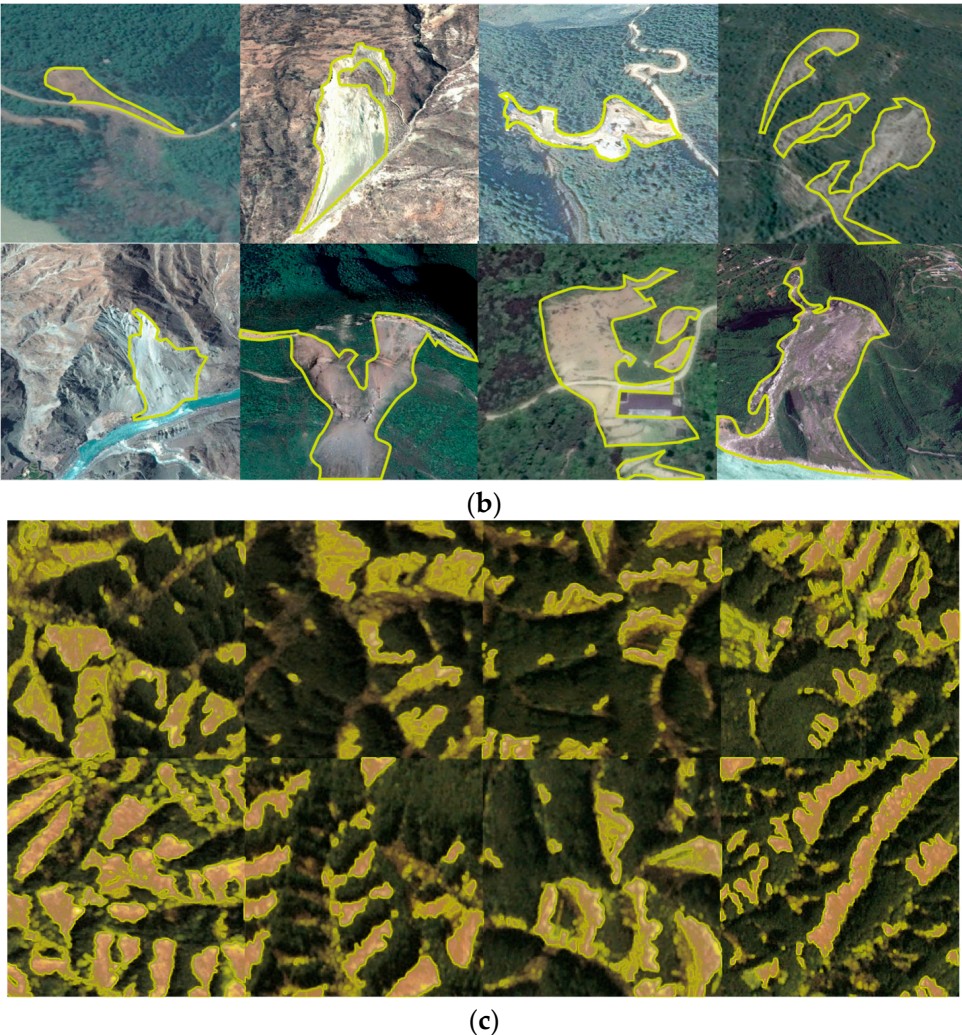

**(b)**

**(c)**

**Figure 2.** Examples of landslide samples. (**a**) Bijie, Guizhou, China. (**b**) Wulong, Chongqing, China. (**c**) Tianshui, Gansu, China.

### 2.2. Methods

The architecture of our model was inspired by the U-Net framework. Recognizing the potential deformation issues that may arise from handling images with inconsistent dimensions, we augmented the network structure with a Pyramid Visual Field Receptor and incorporated a Composite Attention Mechanism, specifically CBAM, to enhance the decoder's capability of restoring low-dimensional data. A schematic of the MsASNet architecture is shown in Figure 3, and more inner core structures of sub-modules are shown in Figure 4.

After preprocessing, the RGB landslide data were resized to dimensions of $512 \times 512$ pixels prior to input into the network. Within the network architecture, the first block is a double-convolutional block whose architecture is shown in Figure 4a. This double-convolutional block consisted of two $3 \times 3$ convolutional layers with a padding size of 1 and a stride of 1. Subsequent to each convolutional layer, a batch normalization layer was applied followed by the Rectified Linear Unit (ReLU) activation function. This double convolutional layer facilitated the transition of feature map dimensions from (B, 3, 512, 512) to (B, 64, 512, 512). Following this, the feature maps traversed through four downsampling blocks. Figure 4b shows the structure of a downsampling block. At the culmination of each downsampling block, the dimensions of the feature maps were halved successively, resulting in dimensions of $256 \times 256$, $128 \times 128$, $64 \times 64$, and $32 \times 32$. These blocks effectively extracted pertinent features from the feature maps, resulting in channel

dimensions of 128, 256, 512, and 512 for the respective downscaling blocks. To further reduce the data dimensions and extract salient features, the last downsampling block was succeeded by another double-convolutional module.

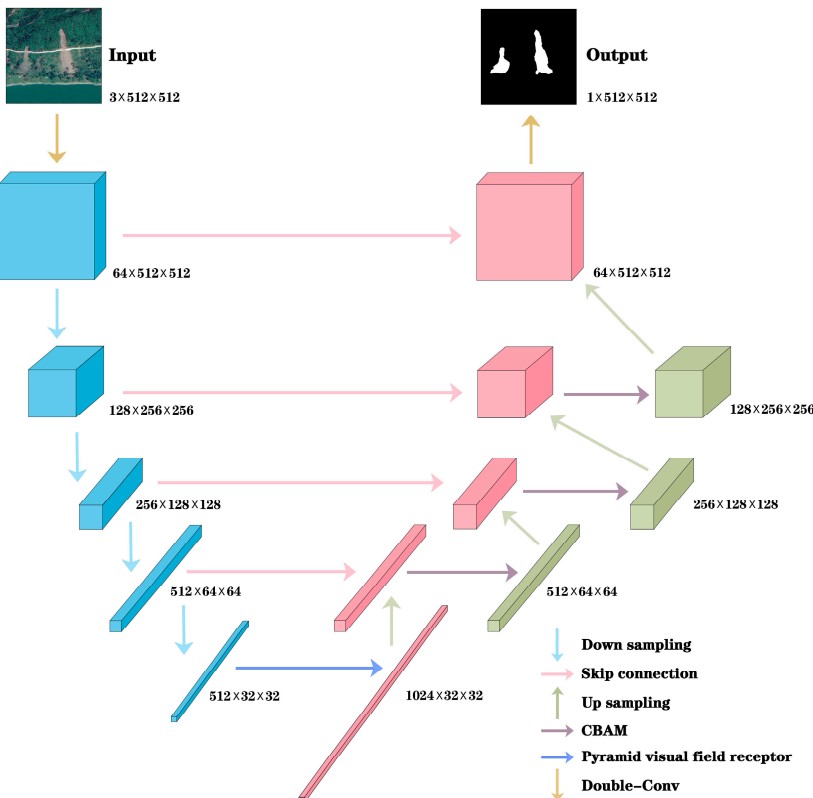

**Figure 3.** Architecture of the MsASNet model.

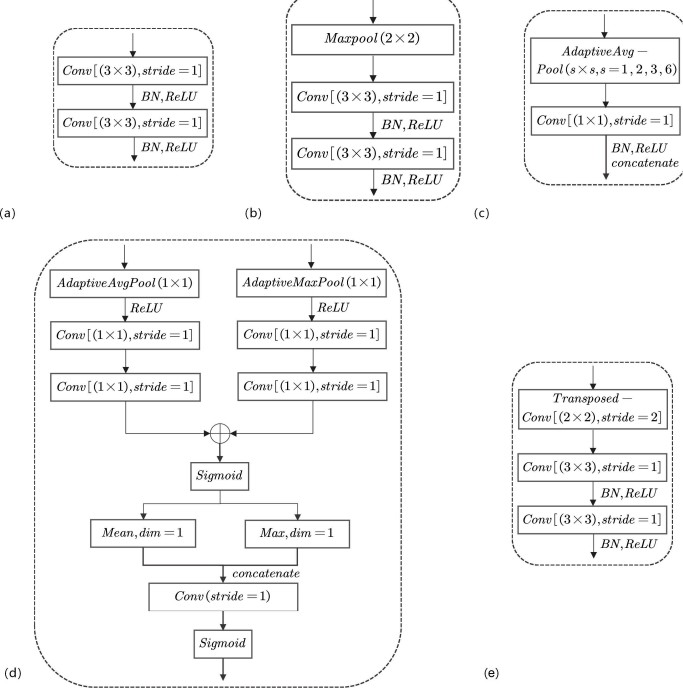

**Figure 4.** Inner core structures of sub-modules. (**a**) is the architecture of double-convolutional block; (**b**) is the architecture of downsampling block; (**c**) is the architecture of the pyramid field of view receptor; (**d**) is the architecture of the CBAM; (**e**) is the architecture of upsampling module.

Due to the use of max pooling for downsampling in the U-Net structure, the image resolution is reduced during this operation, resulting in a certain degree of detail and spatial information loss. Simultaneously, as the network layers increase, the feature maps approach the semantic information in the image, and downsampling may lead to some critical semantic information loss. Therefore, to mitigate the information loss caused by downsampling, we incorporated a dynamic visual receptor based on the pyramid pooling module, known as the Pyramid Field of View Receptor, into the U-Net modules. The structure of the pyramid field of view receptor is shown in Figure 4c. This module obtains features through pooling layers of different scales, and then fuses the extracted features into a vector of fixed dimensions. Finally, it restores the feature maps to an appropriate dimensionality, facilitating subsequent concatenation with the input perceptual feature maps. Upon the application of the Pyramid Visual Field Receptor, the feature map dimensions transition to (B, 1024, 32, 32).

Following the feature extraction phase, the feature maps progressed to the decoding stage. During decoding, the data were transformed from a low-dimensional feature representation back to its original data space, culminating in the generation of the final segmentation results. To harness more informative content from low-dimensional features, a Channel-Attention and Spatial-Attention Module (CBAM) was incorporated preceding each upsampling module. The structure of the CBAM is shown in Figure 4d. CBAM synergistically integrates channel and spatial attention mechanisms, thereby facilitating adaptive adjustments to feature maps based on both channel- and spatial-wise weight allocations. Whereas channel attention emphasizes inter-feature relationships, spatial attention prioritizes the spatial distribution of features. By learning these adaptive weights, the CBAM effectively adapts to diverse feature distributions within the input data, thereby proficiently capturing and leveraging the critical information embedded in the input features to enhance model performance. When integrated in tandem with the decoder module, the CBAM serves to mitigate redundancy within the feature maps. This enables the decoder to concentrate more acutely on salient features beneficial to the task at hand and discern information across varied spatial scales within the feature maps.

When constructing the upsampling module, we utilized transpose convolution. Compared to interpolation-based upsampling operations, transpose convolution possesses parameterized layers, enabling it to alter the dimensions of feature maps through learned features. This allows the model to automatically acquire upsampling patterns suited to the task during the training process. Combining it with CBAM further enhances the upsampling process for landslide segmentation tasks. Through learnable parameters, transpose convolution can adaptively learn upsampling patterns specific to landslide segmentation. Meanwhile, CBAM selectively amplifies important features and diminishes irrelevant information within the feature maps, thereby effectively guiding the upsampling process and improving model performance. The structure of the upsampling module is shown in Figure 4e.

### 2.3. Evaluation Techniques

Throughout this research, pixel-level precision, recall, and F1-score metrics were adopted to quantify the congruence between the outcomes generated by the segmentation algorithms and the corresponding ground truth. Precision, also called the check rate, is the ratio of the correct prediction of the positive sample to the number of positive predictions, which indicates the precision of the prediction. The percentage of accurately anticipated positive samples to the total number of actual positive samples is known as the recall. The F1-score can avoid errors caused by a single identification target to some degree and can measure the model more comprehensively than existing techniques.

Specifically, within the segmented results, pixels correctly classified into the object region were designated as True Positives (TP), whereas those erroneously categorized as background constituted False Negatives (FN). Pixels inaccurately assigned to the object region within the background were denoted as False Positives (FP). Consequently, we pro-

vided a rigorous evaluation framework for assessing the segmentation performance relative to the ground-truth precision and recall, and the F1-score metrics were mathematically expressed in terms of TP, FN, and FP. The formulas used are as follows:

$$\text{Precision} = \frac{\text{TP}}{\text{TP} + \text{FP}} \tag{1}$$

$$\text{Recall} = \frac{\text{TP}}{\text{TP} + \text{FN}} \tag{2}$$

$$\text{F1} = \frac{2 \times \text{Precision} \times \text{Recall}}{\text{Precision} + \text{Recall}} \tag{3}$$

where TP is predicted to be a positive sample, FP is predicted to be a positive sample and a negative sample, and FN is predicted to be a negative and positive sample.

### 2.4. Ablation Study

We propose a novel MsASNet tailored for landslide data segmentation that is predominantly anchored to the U-Net architecture. This framework seamlessly integrates a dynamic visual receptor with a convolutional block attention module, endowing the network with enhanced feature extraction capabilities, efficient information fusion, and superior image segmentation accuracy. To delineate the specific contributions of the individual components within the network architecture, we conducted a comprehensive dissection and subsequent ablation experiments.

During our ablation study, we evaluated the impact of four distinct architectural configurations on network performance. These configurations comprised the foundational U-Net structure, a concatenated framework integrating U-Net with a channel attention module, an amalgamated design combining U-Net with the CBAM module, and an adaptation in which a pyramid-scene parsing module that was embedded between the upsampling and downsampling stages of the U-Net architecture.

In the ablation study, all experimental procedures were conducted on a desktop workstation equipped with specific hardware configurations, including a 13th Generation Intel (R) Core (TM) i9-13900KF CPU operating at 3.0 GHz, complemented by 32 GB of RAM. The computational tasks were further facilitated by an NVIDIA GeForce RTX 4090 GPU processor with 24 GB of dedicated memory running on the Windows 10 professional operating system. During the training phase, optimization was attained using a stochastic gradient descent (SGD) algorithm employing a batch size of eight samples. The initial learning rate was set to 0.0001, and the training regimen spanned 200 epochs.

### 2.5. Comparative Experiments

In this section, we describe a series of experiments conducted to benchmark the performance of the proposed network against several classical segmentation architectures. Specifically, these architectures encompass Fully Convolutional Networks (FCN) [27], which are adept at processing inputs of arbitrary dimensions and generating pixel-level label predictions for equivalent dimensions. Fully convolutional networks (FCNs) and their variants are frequently used in CNN-based models [28]; however, these FCNs use convolutional layers instead of the fully connected layers found in the original CNNs. As a result, they only require convolutional (subsampling or upsampling) operations. Compared with traditional CNNs, FCNs have the following advantages: (1) they avert the disappearance of spatial data, (2) they drastically lower the required computing parameters, and (3) they enhance the capability to represent. Consequently, FCNs are better suited for segmentation tasks. However, they tend to be oblivious to the finer details of an image because of their propensity to disregard pixel relationships [29]. This restriction may make it more difficult for FCNs to efficiently gather global context data [30].

Several variants of FCN that attempt to address this limitation have emerged. For example, SegNet [31] and U-Net [32] are characterized by their encoder–decoder architec-

ture, which facilitates intricate feature reconstruction while maintaining computational efficiency. In these architectures, the spatial dimensions of the object are gradually reduced by the encoder, whereas its details and spatial dimensions are gradually restored by the decoder. Using the skip connection, the decoder obtains data from the encoder portion at the same level as the feature mapping, allowing for tighter localization [33].

The low-dose imaging and limited-angle imaging inpainting Model (LDLAIM) [34] architecture, recognized for its depth and prowess in feature extraction, has demonstrated superiority, particularly in image reconstruction tasks. Furthermore, we examined PSP-Net [35], which leverages pyramid-pooling modules to capture multiscale contextual information; this enhances its semantic segmentation capabilities across diverse receptive fields and augments its proficiency in comprehending scene intricacies.

In addition to the aforementioned conventional segmentation algorithms, we also compared the segmentation performance with two relatively novel landslide-specific segmentation algorithms: ResU-Net-OBIA [36] and DRs-UNet [37]. The ResU-Net-OBIA framework integrates the traditional ResU-Net with object-based image analysis based on four simple rules. Through testing on Sentinel-2 imagery, it was found that this framework can significantly enhance segmentation performance. DRs-UNet incorporates a residual shrinkage building unit into the traditional U-Net network, which effectively reduces the noise level in images using soft thresholding, thereby enhancing the network's feature extraction capability. Table 1 provides a more intuitive overview of these comparative experiments, covering aspects such as number of convolutional layers, kernel size, and activation functions.

**Table 1.** Comparison of parameters of different algorithms.

|  | Number of Convolutional Layers | Kernel Size | Activation Function |
|---|---|---|---|
| FCN | 18 | $3 \times 3, 1 \times 1$ | ReLU |
| SegNet | 26 | $3 \times 3$ | ReLU |
| LDLAIM | 23 | $3 \times 3, 1 \times 1$ | PReLU, ReLU |
| PSPNet | 61 | $7 \times 7, 3 \times 3, 1 \times 1$ | ReLU |
| ResU-Net-OBIA | 15 | $3 \times 3, 1 \times 1$ | ReLU |
| DRs-UNet | 18 | $3 \times 3, 2 \times 2, 1 \times 1$ | Sigmoid, ReLU |

## 3. Results

### 3.1. Results of the Ablation Study

Table 2 presents the quantitative analysis results derived from the four ablation experiments, and red regions in Figure 5 illustrates their respective segmentation results.

**Table 2.** Quantitative analysis of the ablation study.

|  | Precision | Recall | F1 |
|---|---|---|---|
| U-Net | 0.9048 | 0.9200 | 0.9046 |
| U-Net with channel attention | 0.9071 | 0.9214 | 0.9082 |
| U-Net with CBAM | 0.9142 | 0.9380 | 0.9187 |
| U-Net with Pyramid Scene Parsing | 0.9368 | 0.9470 | 0.9385 |
| MsASNet | 0.9510 | 0.9559 | 0.9516 |

In contrasting the outcomes of various ablation experiments, the standalone U-Net model exhibited a competent performance in delineating landslide regions within the remote sensing imagery. However, it demonstrated limitations in accurately segmenting certain images, exhibiting notable discrepancies when compared with ground-truth annotations. The incorporation of a channel attention module into the U-Net architecture yielded marginal quantitative enhancements and visually surpassed the segmentation efficacy of the basic U-Net framework. Substituting the channel attention module with the CBAM

mechanism further augmented network performance, particularly enhancing the segmentation accuracy for intricate landslide categories. Notably, embedding the pyramid-scene parsing module between the upsampling and downsampling phases yielded substantial performance gains, as evidenced by the enhanced segmentation accuracy in the quantitative analyses and superior visual outcomes compared to preceding experiments. The methodology proposed in this study consistently outperformed prior configurations, thereby substantiating the efficacy of leveraging the U-Net architecture augmented with pyramid-scene parsing for efficient contextual feature extraction across multiple scales. Concurrently, cascading the CBAM module during each upsampling stage facilitated adaptive learning, which enabled the network to prioritize regions pivotal to semantic segmentation. This increased both the accuracy and robustness of the segmentation process.

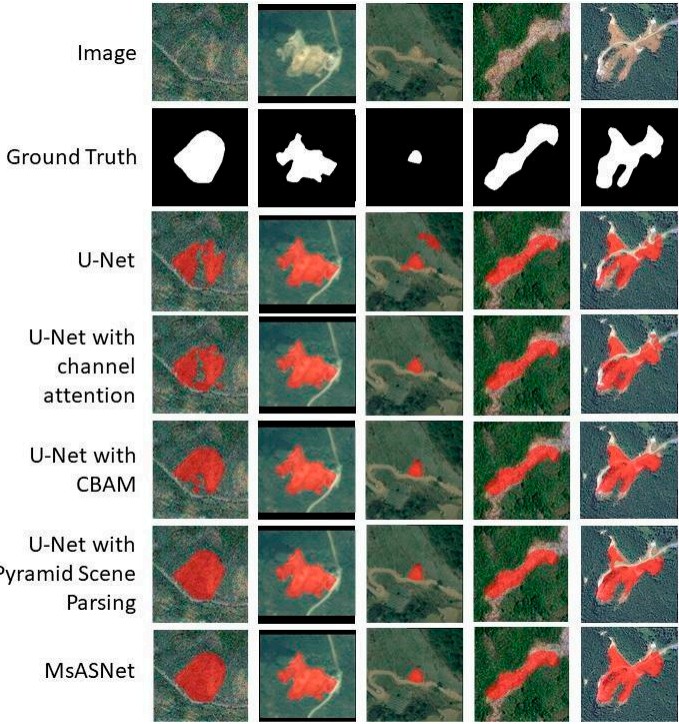

**Figure 5.** Ablation study results. The red regions in these figures indicate the landslide area segmented by various algorithms.

### 3.2. Results of Comparative Experiments

To evaluate the segmentation effectiveness of MsASNet on different types of landslide datasets, we utilized the Chongqing and Tianshui landslide datasets separately. The Chongqing dataset was captured in the southeastern region of Chongqing City, influenced by a subtropical moist monsoon climate, encompassing the Wujiang downstream area and the Wuling Mountain and Dalou Mountain gorge zones in the southeast of Chongqing City. Satellite imagery reveals dense vegetation in this area, with landslide occurrences relatively concentrated rather than scattered across the landscape. Conversely, the region captured in Tianshui's dataset is the loess hill landform under the influence of temperate monsoon climate. The landslide areas in this dataset exhibit characteristics such as large area coverage, complex shapes, and diverse variations, posing significant challenges to the segmentation performance of the network. Hence, we conducted comparative experiments utilizing multiple methods for these two distinct datasets to demonstrate the superior performance of the proposed algorithm.

- Comparative Experiments using Chongqing's landslide dataset

Table 3 presents the quantitative analysis results derived from the four comparative experiments using the Chongqing dataset, and the red regions in Figure 6 illustrates the

selected segmentation results for visual comparison. Upon comprehensive evaluation of the quantitative analyses from the test dataset and the displayed experimental outcomes, both FCN and SegNet tended to misclassify numerous regions, erroneously segmenting areas as landslide zones when they did not correspond to actual landslide regions. Furthermore, when segmenting expansive areas, LDLAIM demonstrated a performance that was generally satisfactory, but not exceptional.

**Table 3.** Quantitative analysis of comparative experiment using Chongqing's landslide dataset.

|  | Precision | Recall | F1 |
| --- | --- | --- | --- |
| FCN | 0.1442 | 0.2863 | 0.2371 |
| SegNet | 0.7919 | 0.3094 | 0.4009 |
| LDLAIM | 0.6777 | 0.6104 | 0.5655 |
| PSPNet | 0.6141 | 0.4922 | 0.4702 |
| ResU-Net-OBIA | 0.6016 | 0.4116 | 0.4260 |
| DRs-UNet | 0.8445 | 0.8583 | 0.8978 |
| MsASNet | 0.9145 | 0.8658 | 0.9010 |

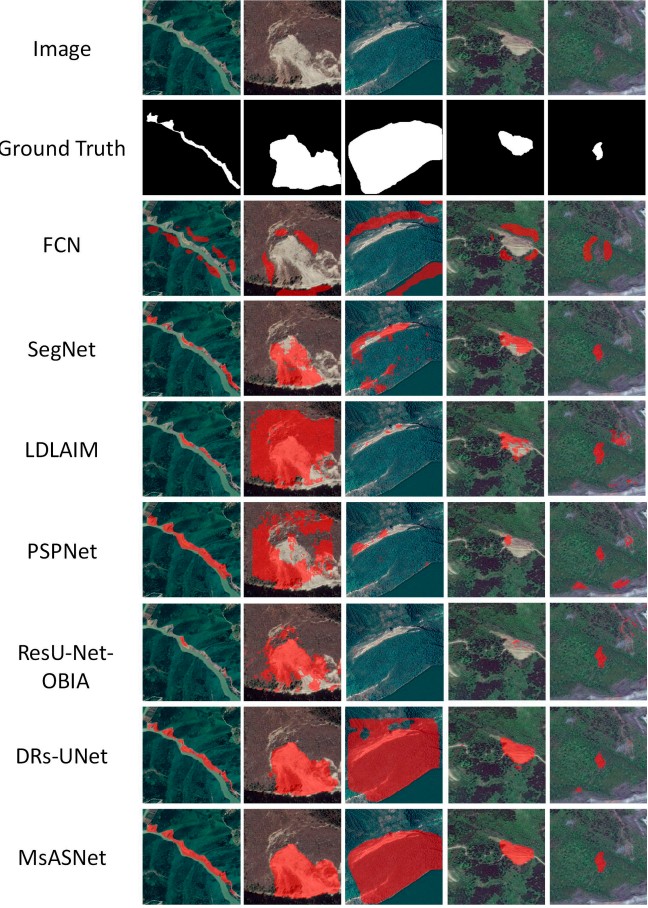

**Figure 6.** Comparative experiment results using Chongqing's landslide dataset. The red regions in these figures indicate the landslide area segmented by various algorithms.

Figure 7 illustrates the variations in loss among several segmentation algorithms during the comparative experiment. From the graph, it is evident that the proposed MsASNet exhibits a rapid decline in loss during the early stages of training. Approximately 70 epochs into the training process, the loss converges to around 0.01, consistently outperforming other comparative algorithms. This observation suggests robust training dynamics and efficient optimization processes inherent in MsASNet.

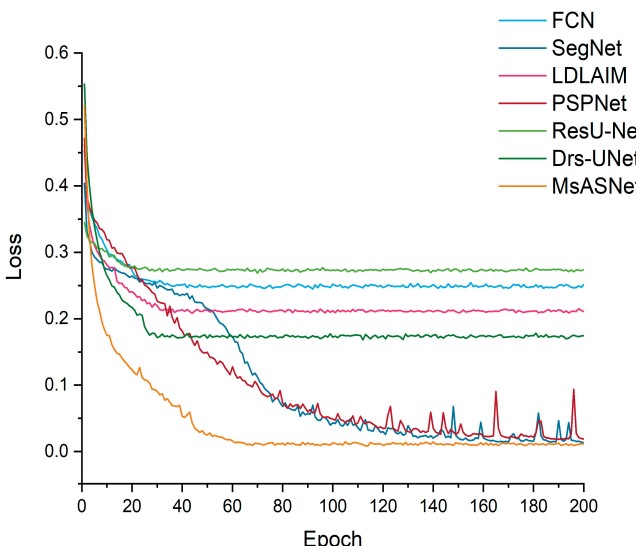

**Figure 7.** Loss functions of different methods used in the training process.

Furthermore, the consistent outperformance of MsASNet in terms of loss convergence underscores its superior fitting capability on the training set compared with other methods. The stable convergence of loss over epochs also implies that the model achieves a desirable level of stability during training, which is crucial for ensuring reliable and consistent performance. This stability in training dynamics contributes to the model's generalization ability, as it indicates that the learned representations are robust and less susceptible to overfitting.

The observed training dynamics and stability of MsASNet not only enhance its performance on the training set but also have significant implications for its generalization to unseen data. A model that exhibits stable training dynamics is more likely to generalize well to new, unseen samples, as it indicates that the learned features are representative of the underlying data distribution. Therefore, the stable convergence of loss in MsASNet during training serves as a positive indicator of its ability to generalize effectively and perform reliably on unseen data, thus enhancing its overall utility and applicability in practical settings.

- Comparative Experiments using Tianshui's landslide dataset

Due to the primary segmentation target of Tianshui's landslide dataset being loess hilly terrain, it exhibits challenges such as large landslide areas, complex shapes of landslide regions, and the presence of multiple landslide areas within individual samples. Consequently, compared to the preceding two landslide datasets, this dataset poses greater segmentation difficulty.

Table 4 shows the quantitative analysis of different comparative experiments conducted using Tianshui's landslide dataset, and the red regions in Figure 8 illustrates the segmentation results of various algorithms on landslide areas within loess hilly terrain.

**Table 4.** Quantitative analysis of the comparative experiments using Tianshui's landslide dataset.

|  | Precision | Recall | F1 | GFLOPs | Params (M) |
|---|---|---|---|---|---|
| FCN | 0.2285 | 0.2976 | 0.3547 | 3.1366 | 11.1774 |
| SegNet | 0.8485 | 0.8619 | 0.8627 | 40.1311 | 22.3610 |
| LDLAIM | 0.8423 | 0.8641 | 0.8991 | 48.6439 | 17.2806 |
| PSPNet | 0.8631 | 0.9018 | 0.8940 | 44.4399 | 46.5823 |
| ResU-Net-OBIA | 0.7861 | 0.6067 | 0.6789 | 651.5271 | 7.8757 |
| DRs-UNet | 0.8238 | 0.8836 | 0.8796 | 11.1131 | 7.9369 |
| MsASNet | 0.9017 | 0.9608 | 0.9030 | 42.7559 | 27.5391 |

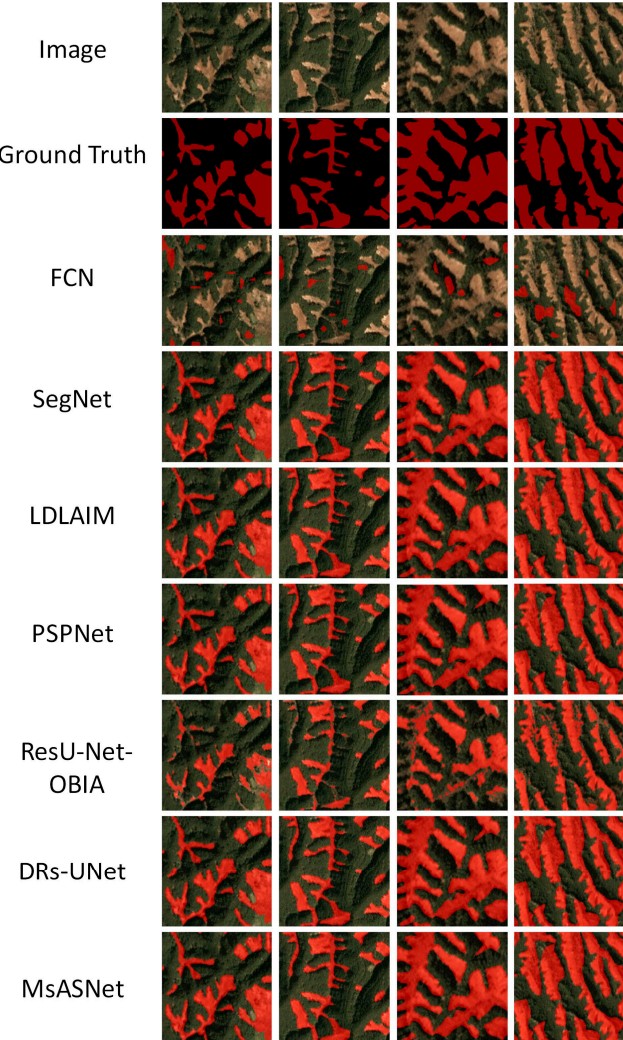

**Figure 8.** Comparative experiment results using Tianshui's landslide dataset. The red regions in these figures indicate the landslide area segmented by various algorithms.

In Table 4, we compared the network complexity and parameter count of our algorithm with several others when processing the dataset. After computation, MsASNet's floating point operations are estimated to be approximately 42.75 G, placing it at a moderate level among these algorithms; its parameter count is approximately 27.53 M, ranking second only to PSPNet in parameter count. Considering other results holistically, and owing to its large number of learnable parameters, MsASNet exhibits favorable performance in terms of accuracy and effectively balances segmentation accuracy with network complexity.

In Figure 8, we have selected representative images from the dataset. The first two images exhibit complex segmented areas with numerous branched landslide regions. These regions, being relatively fine-grained in the images, are susceptible to information loss during feature extraction, thereby presenting a greater segmentation challenge. Conversely, the latter two images feature extensive target regions, posing significant challenges in network architecture design under such circumstances.

By comparing the images in Figure 8, it can be observed that DRs-UNet, PSPNet, and SegNet tend to further refine the morphology of small, branched regions during segmentation. Conversely, LDLAIM may generate some voids within the target area when segmenting larger landslide regions. The proposed algorithm in this study demonstrates proficiency in addressing both scenarios, effectively preserving the shape of landslide areas. This observation is supported by the quantitative analysis presented in Table 4.

## 4. Discussion

Previous studies have explored the use of deep learning for landslide identification. Ghorbanzadeh et al. used ResU-Net and OBIA for landslide identification in multitemporal Sentinel-2 images, and their results showed an F1-score of 84.03% and a mIoU value of 72.49% [36]. Liu et al. [38] suggested employing ResU-Net to identify earthquake-related landslides in the Jiuzhaigou Valley Scenic and Historic Interest Area in the Sichuan Province, China. Their results demonstrated a 93.3% F1-score and an 87.5% mIoU value. The prediction capabilities of U-Net and SegNet differ; the former converges quickly and produces accurate predictions for small sample sets [39]. Achieving favorable prediction results with SegNet is more likely in scenarios with a large sample size; however, importantly, the accuracy of the labels plays a pivotal role in the overall effectiveness of SegNet [40]. The utilization of pyramid visual field receptors in MsASNet post downsampling effectively mitigates the semantic loss incurred during downsampling. Concurrently, the pyramid visual field receptor facilitates feature acquisition through pooling layers at different scales, followed by the fusion of extracted features into a fixed-dimensional vector. It is precisely due to this dynamic feature fusion mechanism that the loss incurred during the propagation of features across networks for images of varying sizes is minimized. In contrast, the MsAS-Net framework proposed in this study demonstrated superior performance metrics across the precision, recall, and F1-score evaluations, outpacing the aforementioned architectures by significant margins. In the comparative experiments based on the Chongqing landslide dataset, MsASNet exhibited a significant advantage over the second-ranked DRsU-Net model, with increases of 8.3%, 0.8%, and 0.3% in precision, recall, and F1-score values, respectively. Furthermore, when dealing with a dataset characterized by fewer samples and greater segmentation difficulty, such as the Tianshui loess hilly landslide dataset, MsASNet demonstrated significantly greater advantages in quantitative analysis and exhibited superior visual performance. Through data analysis, it outperformed the second-ranked PSPNet, with increases of 4.5%, 6.5%, and 1.0% in precision, recall, and F1-score values, respectively. Additionally, from a qualitative perspective, MsASNet's segmentation results exhibited closer conformity with ground truth annotations, further underscoring its efficacy and robustness in landslide segmentation tasks. However, it is undeniable that the network still exhibits certain shortcomings that require improvement. Upon examining segmentation results across a broader spectrum of samples, it becomes evident that the proposed network demonstrates room for enhancement in handling contour details, particularly in regions with sharp ground-truth edges. Additionally, the study relies on higher-quality remote sensing data to facilitate improved segmentation and extraction of feature areas. Especially for the extraction of smaller potential landslides, the optical image data should have better resolution. The conclusions drawn from this study provide research ideas for the study of soil properties in different areas, and further exploration of the applicability of the model to more areas is subject to further research.

## 5. Conclusions

In this study, to address the current dilemma in the field of geological disaster recognition based on the current excellent algorithms in the field of deep learning, we improved the semantic segmentation network, designed a targeted landslide disaster feature extraction and recognition model, MsASNet, and conducted ablation tests on the model using Bijie's landslide dataset from Guizhou Province. The experimental results show that the maximum segmentation overall accuracy of MsASNet in this dataset reached 95.10%, and the maximum F1-score reached 95.16%. In addition, we used the landslide disaster datasets from the mountainous area of southwest Chongqing and the loess hilly landslide disaster in Gansu Tianshui to compare other segmentation algorithms. Findings from the experiments reveal that, compared with other segmentation algorithms, the model accurately identified and segmented the range of landslides in the image. Quantitative analysis results substantiate that MsASNet provided marked performance enhancement across multiple evaluation metrics when juxtaposed against other classical networks. Upon

comprehensive comparison of the segmentation results of MsASNet across three distinct datasets, we observed that the algorithm tends to excessively smooth sharp edges of contour delineations, particularly evident in landslide data characterized by sharply defined edges. This phenomenon results in inaccuracies in the segmentation of landslide areas. Consequently, in future endeavors, further optimization of the model architecture and the development of sophisticated loss functions to facilitate training processes are warranted. These efforts aim to enhance network adaptability to diverse landslide segmentation tasks across varying geographic regions. However, it is undeniable that the model still exhibits high segmentation accuracy when confronted with scenarios of limited sample quantities and the presence of multiple complex segmentation regions within each sample. Although MsASNet achieves higher segmentation accuracy, we observed that during training, it consumes more GPU memory compared to several other comparative algorithms. This is attributed to its larger model parameters, indicating that the network may face challenges when resources are limited. Consequently, on the one hand, it is necessary to continue to expand the data sources and scale, and to improve the accuracy of dataset annotation so as to continue to support research on the structure of the semantic segmentation model for landslide hazards in remotely sensed images; on the other hand, future endeavors will require refining the network to augment its computational efficiency. There is a need to further bolster MsASNet's capabilities, as this improvement may ensure its efficacy in addressing landslides across complex or variable geographical and climatic conditions.

**Author Contributions:** Conceptualization, N.Z. and Z.Z.; methodology, N.Z.; software, Z.Z.; validation, W.C., S.W. and J.H.; formal analysis, N.Z.; investigation, N.Z.; resources, W.C. and S.W.; data curation, Z.Z.; writing—original draft preparation, N.Z.; writing—review and editing, Z.Z.; visualization, N.Z.; supervision, Z.Z. and J.H.; project administration, W.C. and S.W.; funding acquisition, W.C. and S.W. All authors have read and agreed to the published version of the manuscript.

**Funding:** This research was funded by the National Natural Science Foundation of China (grant number 42205146), the National Key Research and Development Program of China (grant number 2022YFF0711703), and the Zhongke Bengbu Technology Transfer Center Project (grant number ZKBB202201).

**Data Availability Statement:** The data presented in this study are available on request from the corresponding author. The data are not publicly available due to privacy reason.

**Acknowledgments:** Thanks for the V&V support by the Institutional Center for Shared Technologies and Facilities of the Institute of Nuclear Energy Safety Technology, Hefei Institutes of Physical Science, Chinese Academy of Sciences.

**Conflicts of Interest:** The authors declare no conflicts of interest.

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
