# Peer review of "A Multiscale Attention Segment Network-Based Semantic Segmentation Model for Landslide Remote Sensing Images"

_remotesensing, doi:10.3390/rs16101712_

Round 1
Reviewer 1 Report (Previous Reviewer 3)
Comments and Suggestions for Authors
1. This paper does not sufficiently demonstrate a significant advancement over existing models. The comparison with other established models like U-Net and SegNet lacks depth, and the incremental improvements reported do not clearly justify the need for a new model. Further, the manuscript fails to address how this model compares in contexts other than the ones tested.
2. The manuscript is poorly organized and lacks clarity in its explanations, making it difficult for readers to follow the development and evaluation of the proposed model. Key components of the methodology are described ambiguously, which casts doubt on the reproducibility of the results.
3. The introduction lacks a thorough review of the existing literature. It does not adequately contextualize the proposed model within the broader field of landslide detection or discuss how this model addresses unmet needs.
4. The description of the MsASNet model is vague about certain crucial aspects, such as the configuration of layers and the specific enhancements made over similar models. This lack of detail might prevent others from replicating the study or understanding the theoretical underpinnings of the model's design.
In summary, while the topic is of significant interest and potential impact, the manuscript as currently presented lacks the scientific rigor, clarity, and detail needed to meet the publication standards. Significant revisions and more robust experimental validations are needed before it can be reconsidered for publication.
N/A.
Author Response
Please see the attachment.

Reviewer 2 Report (Previous Reviewer 5)
Comments and Suggestions for Authors
The manuscript presents a novel deep learning model, MsASNet, designed for semantic segmentation of landslide remote sensing images. By incorporating multiscale attention mechanisms and an encoder-decoder structure, the model aims to improve the precision of landslide detection in complex terrains.
The manuscript thoroughly introduces the MsASNet model, detailing its architecture and the rationale behind its design. However, a deeper comparison with existing models (e.g., U-Net, SegNet, and FCN) in terms of architecture differences, operational efficiency, and practical applicability would enrich the readers' understanding. It's recommended to provide a clearer, tabulated comparison that highlights the unique features and advantages of MsASNet over these established models.
The manuscript highlights the model's potential for practical application in landslide detection and disaster prevention. Further discussion on the model's integration with existing disaster management systems, potential challenges in real-world deployment, and strategies to overcome these challenges would provide a more comprehensive view of its applicability.
Given the complex architecture of MsASNet and its superior performance, it would be beneficial to discuss its computational efficiency and resource requirements in detail. Understanding the trade-offs between accuracy and computational demand is crucial for deploying such models in operational settings, especially in regions with limited computing resources.
I wish that my comment would be helpful in improving the quality of this research.
Thank you.
Author Response
Please see the attachment.

Reviewer 3 Report (Previous Reviewer 4)
Comments and Suggestions for Authors
The article presents a study on enhancing landslide detection from remote sensing images using a novel deep-learning model called MsASNet. This model, which utilizes a multiscale attention segment network, innovatively addresses the semantic segmentation process—a critical component for geological hazard identification. By incorporating an encoder-decoder architecture with a channel attention mechanism, MsASNet aims to overcome the limitations of previous models, particularly in improving the accuracy of landslide boundary detection and feature extraction. The methodology could benefit from a more detailed justification of its architectural choices, explicitly linking them to the unique challenges of landslide detection. The practical implications of this research are significant, especially in disaster management, where the model's capability for real-time processing can substantially aid in early warning systems, potentially mitigating the impact of landslides. While MsASNet demonstrates superior precision, recall, and F1-score metrics performance, further discussion is needed on optimizing model complexity to enhance computational efficiency and handle diverse geographic conditions. The article notes the model's tendency to smooth sharp contour edges, a limitation that could lead to inaccuracies, suggesting a need for refinement to improve performance in such scenarios. Expanding the discussion on the model's generalizability and simplifying technical jargon could make the findings more accessible and relatable to a broader audience. Rigorous validation against other state-of-the-art models suggests that MsASNet could set a new benchmark in remote sensing applications for landslide detection.
Line 329 has a seperation mark left "32 GB of RAM. The compu-tational"
Line 329 -330, I believe the authors used RTX 4090, not GTX.
Line 332 has a separation mark "attained using a stochastic gra-dient descent (SGD)"
Figure 6. Most of the competition examples seem not fit for the task, therefore comparison seems a bit odd.
Author Response
Please see the attachment.

Reviewer 4 Report (Previous Reviewer 2)
Comments and Suggestions for Authors
Zhou and colleagues propose the MsASNet model, a novel strategy aimed at improving the prediction of landslide disasters. This model integrates sophisticated features such as channel attention, CBAM, and Pyramid Scene Parsing with the established Unet framework. The introduction of MsASNet offers notable advancements in performance across both tested and newly introduced datasets, highlighting its potential in enhancing predictive accuracies. Despite these advancements, the manuscript would benefit from several revisions to fully realize its publication potential. These enhancements would ensure a comprehensive evaluation and presentation of the research findings.
First, the employment of precision, recall, and F1 score as the primary metrics, standard within segmentation tasks, requires a more detailed exploration concerning their benefits and drawbacks specific to landslide disaster predictions. The significant impact of completely overlooking a landslide, compared to minor inaccuracies in prediction, suggests that alternative metrics from object detection fields might provide a more nuanced assessment of MsASNet’s capabilities.
Second, the manuscript should offer a clearer explanation of the approach used to divide the dataset into training and testing subsets, including the number of samples in each. Such transparency is crucial for the reproducibility and thorough understanding of the study's methodology and results.
Third, the paper contains several citation errors that need rectification, such as the misplaced reference noted on line 373 marked by “Error! Reference source not found.” Correcting these errors will enhance the paper's credibility and readability.
Fourth, the comparison of MsASNet against baseline methods in Table 3 suggests a need for stronger benchmarks. Given the demonstrated efficiency of the vanilla UNet in Figure 5, its inclusion in Table 3 as a comparison point would strengthen the argument if no superior methods are available.
Incorporating these suggestions would not only clarify and enrich the manuscript but also reinforce its contribution to advancing landslide prediction methodologies, paving the way for its acceptance in a scholarly publication.
Author Response
Please see the attachment.

Reviewer 5 Report (Previous Reviewer 1)
Comments and Suggestions for Authors
After careful consideration of the feedback received, the manuscript has been revised in accordance with the recommendations.
Author Response
Please see the attachment.

This manuscript is a resubmission of an earlier submission. The following is a list of the peer review reports and author responses from that submission.
Round 1
Reviewer 1 Report
Comments and Suggestions for Authors
The main question addressed in this manuscript is the development of a Semantic Segmentation model for landslide remote sensing using MsASNet. This topic is original and highly relevant to the field of remote sensing.
While the manuscript is generally clear, it could benefit from a more structured presentation following the IMRAD format. In particular, paragraph 2.2 should be renamed to "Methods" instead of MsASNet.
Compared to other published materials, this manuscript stands out for its extensive utilization of a large landslide database in a neural network model.
Minor improvements needed include:
Clarification of the tripleSat images in the materials section, including their specifications.
Specification of the DEM used, including its source, spatial resolution, and accuracy.
Clarification of the term "landslide images" on line 147, specifying whether it refers to spatial resolution.
Explanation of the sources of images obtained from Google Earth, as it aggregates images from providers such as MAXAR and Digital Globe.
Revision of the term "DEM map" to "elevation map," as DEM represents a model rather than a map.
Overall, the manuscript is scientifically sound, with an appropriately designed experimental setup to test the hypothesis. The results are reproducible based on the detailed methods provided.
The illustrations effectively convey the results and are easily interpretable. Additionally, the references are up-to-date and relevant.
Reviewer 2 Report
Comments and Suggestions for Authors
Zhou and colleagues introduce the MsASNet, an approach designed to enhance landslide disaster predictions by incorporating elements like channel attention, CBAM, and Pyramid Scene Parsing into the foundational Unet architecture. While the proposed MsASNet demonstrates promising performance improvements, several areas require substantial refinement for the work to be considered for publication.
1. The manuscript needs to more clearly articulate the novel contributions of the MsASNet model. The incorporation of established image segmentation techniques into Unet, while potentially beneficial, does not in itself constitute a clear innovation. The authors must elucidate the specific advancements their approach offers, especially in the context of landslide disaster applications, and explain the rationale behind the inclusion of each new module.
2. The effectiveness of channel attention within the model, as indicated by the ablation study, does not show a significant impact. The authors are advised to provide a more detailed justification for retaining this feature in the final iteration of the model.
3. The selection of comparative models presented in Table 2 raises questions regarding the benchmarking process. Given the extensive exploration of Unet and its variants across various domains, it is critical for the authors to benchmark MsASNet against current state-of-the-art methodologies to demonstrate its relevance and superiority conclusively. The current selection of models for comparison appears inadequate for establishing the proposed method's significance.
4. There are several inaccuracies in the referencing of Figures and Tables within the document. These errors disrupt the reader's ability to follow the narrative and assess the evidence provided. A thorough review and correction of these references are imperative.
5. The presentation of Figure 3 is problematic, with its small size and indistinct color differentiation making it difficult for readers to interpret the data accurately. Enhancements to the figure's clarity and legibility are necessary to effectively convey the intended information.
Addressing these concerns with comprehensive revisions will be crucial in clarifying the contributions of the MsASNet and its applicability to landslide disaster prediction, thereby strengthening the manuscript for potential publication.
Reviewer 3 Report
Comments and Suggestions for Authors
1. Introduction and Motivation: While the topic is of interest, the introduction lacks a compelling argument for the novelty and urgency of the research. A clearer articulation of the gap in current knowledge and how this work addresses it is needed.
2. Literature Review: The literature review does not sufficiently differentiate the proposed method from existing work. A more detailed comparison highlighting unique contributions is necessary.
3. Methodology Clarity: The description of the proposed MsASNet model lacks sufficient detail for replication and understanding. Clarifying the model's architecture, parameters, and design choices is essential.
4. Experimental Design: The selection of datasets and the rationale behind them are not adequately justified. Details on the preprocessing steps, data augmentation, and validation methods need to be more thoroughly explained.
5. Results and Analysis: The presentation of results lacks depth, particularly in discussing the model's performance in challenging scenarios. A more comprehensive analysis, including failure cases and limitations, is required.
6. Discussion on Novelty and Impact: The discussion does not convincingly argue the practical and theoretical implications of the findings. A more critical examination of the model's innovation and its contribution to the field is necessary.
7. Conclusion and Future Work: The conclusion is overly optimistic without acknowledging the study's limitations. Future work should be framed within the context of these limitations and potential areas for improvement.
Overall Contribution and Originality: Despite the potential application of MsASNet in landslide detection, the paper does not sufficiently advance the state of the art or contribute novel insights to the field.
Comments on the Quality of English LanguageN/A.
Reviewer 4 Report
Comments and Suggestions for Authors
The article presents MsASNet, a new semantic segmentation model for landslide detection using remote sensing images. It addresses the limitations of existing methods by introducing a multiscale attention segment network, which improves feature extraction and landslide boundary delineation. The model demonstrated high accuracy in tests, suggesting it is applicable in geological disaster prevention and control. The study illustrates significant advancements in remote sensing technology for landslide identification, providing a promising tool for timely and accurate geological hazard analysis.
Comments on the Quality of English LanguageEnglish is mainly good with some unclarities or wording. The text is generally well-composed, but there are occasional issues with phrasing or grammar that could be improved for better clarity and academic tone.
Line 48: "With the emergence and as Earth observation technology evolves, the significance of remote sensing technology in identification is growing and providing early warning for large-scale natural disasters; this is because of its timeliness, large coverage area, and rich information."
Suggested Improvement: "As Earth observation technology evolves, the significance of remote sensing in identification has grown, providing early warnings for large-scale natural disasters due to its timeliness, extensive coverage area, and detailed information."
The original sentence is awkwardly structured and unclear. The suggested improvement clarifies the subject and aligns the reasons for the significance of remote sensing technology more clearly.
Reviewer 5 Report
Comments and Suggestions for Authors
The introduction provides a comprehensive overview of landslides, their prevalence, and the need for efficient monitoring methods, but authors should also highlight the global impact of landslides to underscore the broader significance of the research.
While the background on landslide monitoring methods is well-articulated, emphasizing the limitations of traditional approaches, it is a good idea to briefly discuss recent advancements in remote sensing technologies and their role in disaster management. Also, authors should provide examples or case studies illustrating the challenges faced with manual surveys.
In the end of the introduction, it is a good idea to include a forward-looking statement, suggesting potential future developments or applications of the MsASNet model.
Provide a concise explanation of why the Pyramid Visual Field Receptor and CBAM were selected in the MsASNet architecture, highlighting their specific advantages in the context of landslide segmentation.
Briefly mention why the chosen hardware configuration and training parameters were selected, highlighting any considerations that led to these choices.
Add a brief rationale for the selection of the Bijie's and Chongqing's datasets, emphasizing their relevance to landslide detection and how they contribute to the diversity of the study.u
It is a good idea to include additional visual aids or qualitative assessments to elucidate specific instances where MsASNet excels compared to other models and provide side-by-side visual comparisons in different scenarios to reinforce the superior performance of MsASNet.
Extend the discussion on the training dynamics, explaining the implications of the observed loss convergence in terms of model stability and generalization to unseen data.
Highlight the competitive advantage of MsASNet by showcasing its superior performance metrics (Precision, Recall, F1-score) compared to these previous architectures (Ghorbanzadeh et al. and Liu et al.)
I wish that my comment would be helpful in improving the quality of this research.
Thank you.